# An Observation-Based Methodology and Application for Future

# Atmosphere Secondary Pollution Control via an Atmospheric

# **Oxidation Capacity Path Tracing Approach**

4 5 6

1

2

3

Ke Yue <sup>a,b</sup>, Yulong Yan <sup>a,b,\*</sup>, Yueyuan Niu <sup>c</sup>, Jiaqi Dong <sup>a,b</sup>, Chao Yang <sup>d</sup>, Yongqian Zhou <sup>a,b</sup>, Danning Wang <sup>a,b</sup>, Junjie Li <sup>a,b</sup>, Zhen Li <sup>a,b</sup>, Lin Peng <sup>a,b,\*</sup>

7 8

9

10

11

12 13 <sup>a</sup> Engineering Research Center of Clean and Low-carbon Technology for Intelligent Transportation, Ministry of Education, School of Environment, Beijing Jiaotong University, Beijing 100044, China

b School of Environment, Beijing Jiaotong University, Beijing 100044, China
 c Flight College, Shandong University of Aeronautics, Binzhou, Shandong, 256600,
 China
 d Shanxi Climate Center, Taiyuan, Shanxi 030006, China

141516

\* Corresponding author, E-mail: <a href="mailto:yanyulong@bjtu.edu.cn">yanyulong@bjtu.edu.cn</a>

17 18

19

\* Corresponding author, E-mail: penglin6611@163.com

# Graphical abstract

#### **Abstract**

22

23 As China's emission reduction efforts enter a plateau phase due to the slow decline of secondary pollutants, existing control strategies face diminishing returns. Atmospheric 24 Oxidation Capacity (AOC), a key driver of secondary pollutant formation, represents 25 a critical yet underutilized target for more effective control. The Atmospheric 26 27 Oxidation Capacity Path Tracing (AOCPT) approach was proposed in this study. This approach quantitatively traces AOC to its precursors and sources, thereby facilitating 28 29 the coordinated control of secondary pollution, by integrating three modules: a 30 Radiation Equivalent Oxidation Capacity (REOC) method to quantify precursor species contributions, a Relative Incremental AOC (RIA) metric derived from a 31 coupled box-receptor model to assess source impacts, and a modified source 32 33 apportionment technique to resolve the respective contributions of both precursor 34 species and sources to AOC. Successfully validated in a field study in Changzhi, China, AOCPT identified industrial processes (26.8%) and diesel vehicle emissions 35 (24.1%) as the dominant AOC sources in a case city, driven largely by their 36 trans-2-butene emissions (49.3% and 20.6% of total trans-2-butene, respectively). 37 38 Crucially, secondary organic aerosols (SOA) were inadvertently enhanced by ozone (O<sub>3</sub>)-targeted abatement, an AOC-centric strategy enables the co-mitigation of both 39 40 pollutants. By enabling the precise regulation of AOC through direct quantification of precursor and source roles, the AOCPT approach facilitates the synergistic control of 41 secondary pollutants. It provides a robust technical pathway and theoretical 42 foundation to overcome current challenges in air quality management. 43 Key words: atmospheric oxidative capacity; secondary pollutants control; ozone;

#### 1 Introduction

self- reactions; methodological; observational study

Atmospheric Oxidation Capacity (AOC), which comprises reactive oxidants such as hydroxyl radicals (OH·), ozone (O<sub>3</sub>), and nitrate radicals (NO<sub>3</sub>·), etc. acts as the chemical engine that governs the transformation efficiency of precursors such as volatile organic compounds (VOCs), nitrogen oxides (NO<sub>x</sub>) and sulfur dioxide (SO<sub>2</sub>) into secondary pollutants including O<sub>3</sub>, secondary organic aerosols (SOA), sulfates, and nitrates, etc. (Yu et al., 2022). Modulation of AOC, which directly affect the secondary pollutant formation potential.

54 55 56

64

47

49

53

> In recent years, the Chinese government has achieved substantial reductions in primary pollutant emissions, yet air quality challenges stemming from secondary pollutants remain unaddressed. With the implementation of "the Air Pollution Prevention and Control Action Plan" and "the Three-Year Action Plan on Defending the Blue Sky", the NO<sub>x</sub> and CO, etc. concentrations of China were reduced by 31% and 33.3% respectively in 2024 compared to 2018 (Mep, 2024). However, the environmental and health risks caused by secondary pollutants remain a major concern. In 2024, the concentrations of O<sub>3</sub>-8H in the Beijing-Tianjin-Hebei and its surrounding areas, the Yangtze River Delta, and the Fenwei Plain, which are the main

population, economic, and industrial clusters in China, were still as high as 187 μg/m<sup>3</sup>, 169 μg/m<sup>3</sup>, and 182 μg/m<sup>3</sup>, respectively (Mep, 2024). From 2013 to 2020, there was a significant decrease in primary organic aerosols (POA) in China, but the decrease in secondary organic aerosols (SOA) was relatively small (Chen et al., 2024). Anthropogenic sources have dominant contributors to SOA in central and eastern China (Hu et al., 2017). Furthermore, the high contribution of secondary pollutants during haze events in China has been shown to be directly correlated with strong AOC (Huang et al., 2014; Wang et al., 2022a). Not only the photochemical formation of O<sub>3</sub> is driven by the AOC, but a contribution of up to 80% to fine particulate matter pollution was also attributed to the AOC-driven formation of SOA (Huang et al., 2014; Zhao et al., 2020). Meanwhile, previous studies have shown that environmental issues caused by secondary pollutants result in increased mortality from respiratory and cardiovascular diseases (Zhang et al., 2022), a consequence that further entails socioeconomic losses including medical costs, productivity declines, and analogous economic burdens (Xie et al., 2017). It can be seen that current precursor emission control policies have failed to substantially mitigate the environmental, health, and economic impacts stemming from secondary pollution.

81 82 83

84

85 86

87

88 89

97

105

107108

70

72

75

78

Studies have indicated that refined emission control strategies for primary pollutants, which implement source-specific mitigation measures, demonstrate greater environmental efficacy than broad reduction policies. Wu and Xie et al. (2017) achieved refined control of O<sub>3</sub> pollution by constructing a speciated emission inventory and analyzing the emission contributions of different species in different emission sources (Wu and Xie, 2017). Ding et al. (2022) highlighted the necessity of simultaneous VOCs and NOx emission source control for relieve secondary pollution (Ding et al., 2022). Liang et al. (2024) employed RO<sub>2</sub>· radicals to identify critical VOCs species that significantly influence O<sub>3</sub> generation, further highlighting the mainly emission source contributors of ozone formation (Liang et al., 2024). Most of these studies neglected the importance of AOC. However, due to the uncertainty of emission inventories and the complex chemical mechanisms of secondary pollutant formation (Reis et al., 2009). It is necessitating prioritized investigation of AOC, which the principal driver of secondary pollution generation (Li et al., 2024). Studies demonstrate elevated SOA concentrations in densely populated regions (e.g., North China Plain, Yangtze River Delta), where SOA strongly correlates with precursor pollutants and enhanced AOC (Chen et al., 2024). However, North China Plain maintains persistently high SOA levels despite substantial precursor pollutants emission reductions. Especially the challenge of coordinated secondary pollution control became evident, as previous studies showed that attempts to control O<sub>3</sub> could unexpectedly increase SOA levels due to the non-linear relationship with their precursors (Niu et al., 2024; Lyu et al., 2022). Therefore, emission control that focusing solely on single secondary pollutant impacts while overlooking source contributions to AOC, may lead to deviations in the formulation of emission reduction strategies (Le et al., 2020; Galbally, 2007). Culminating in substantial precursor pollutants reductions yet persistent secondary pollution severity. Existing studies of

AOC predominantly focus on chemical mechanisms and radical interactions (Yu et al.,

2022; Mochida et al., 2003), with limited exploration of emission-driven oxidation

dynamics. Compared with relying on source analysis and control strategies for

individual secondary pollutants, direct traceability analysis of AOC may be more

representative and regulatory efficacy for secondary pollution control.

114

Herein, we developed and applied an Atmospheric Oxidation Capacity Pathway 115 116 Tracing (AOCPT) approach to advance secondary pollution control. This approach 117 quantifies standardized precursor impacts on atmospheric oxidation capacity (AOC) using a metric, the Radiation Equivalent Oxidation Capacity (REOC). It then directly 118 links emission sources to AOC by defining the Relative Incremental Atmospheric 119 120 Oxidation Capacity (RIA) through coupled observation box model (OBM) - positive 121 matrix factorization model (PMF) analysis. Finally, a refined source apportionment 122 method was proposed to quantitatively resolve the contributions of both specific 123 precursors and emission sources to total AOC. By applying this approach, our study comprehensively evaluates emission source contributions and explicitly traces their 124 critical formation pathways through AOC, providing a scientific basis for the 125

127 128

#### 2 Methodology

## 2.1 Site description and data collection

To test and apply the proposed methodology for secondary pollution control, a

synergistic control of both O<sub>3</sub> and SOA through targeted AOC regulation.

- continuous field campaign was carried out in Changzhi, a typical industrial city in
- China, from August 21 to 28, 2024. A detailed description of the study site's industrial
- characteristics and the specific sampling locations is provided in Supplementary
- Material Text S1. The Environmental Monitoring Station of Changzhi provided
- hourly data for key trace gases, including O<sub>3</sub>, NO, NO<sub>2</sub>, and CO, as well as for
- meteorological parameters (temperature, relative humidity, and atmospheric pressure,
- etc.).

- A total of 81 VOCs species were continuously sampled at 2 hours sampling frequency
- by using 3 L stainless steel canisters (SUMMA canister, Entech Instruments Inc.,
- California, USA), and were then stored at indoor temperature and analyzed within a
- week of sampling. The ambient samples were analyzed using a pre-concentrator
- (Entech 7200A Instruments Inc., USA) coupled with a gas chromatograph-mass
- selective detector/flame ionization detector (GC-MSD/FID, Agilent
- 7890GC/5975MSD/FID, USA). The detailed samplings and analysis steps were
- presented in Text S2.

## 148 2.2 Atmospheric oxidation capacity path tracing approach (AOCPT)

#### 149 Part 1: Calculation of the initial concentration of VOCs

In this study, initial VOCs (InVOCs) were considered as VOCs directly emitted from

sources, calculated by Eq. (1) (Wang et al., 2022b).

$$[VOC_i]_{In} = [VOC_i]_M + exp(k_i[OH]\Delta t)$$
(1)

where  $[VOC_i]_{In}$  and  $[VOC_i]_M$  were initial VOCs and measured VOCs concentration for specie i, respectively (ppbv);  $k_i$  denotes reaction rate constant between  $VOC_i$  and OH· radicals (cm³·molecule-¹·s-¹); [OH] represents the OH· radicals concentration (molecule·cm-³), which was simulated by box model; and  $\Delta t$  represents the photochemical age or time that VOC<sub>i</sub>'s reaction with OH· radicals (s), detail information for  $k_i$  and  $\Delta t$  calculation are presented in Text S3.

161 162

## Part 2: Quantifying radical-specific contributions to AOC: A novel unified

## approach via OBM and radical cycling analysis

- Step 1: The AOC and the formation of secondary pollution O<sub>3</sub> was simulated using
- the Master Chemical Mechanism (MCM) in Framework for 0-D Atmosphere
- Modeling (F0AM) software (Jenkin et al., 2015; Wolfe et al., 2016). This open-source,
- zero-dimensional (0-D) box model has been widely used (Nault et al., 2024), and a
- detailed introduction to its application can be found in our previous research (Niu et
- al., 2024).

- The AOC could be represented by the sum of the reaction rates of VOCs, CO, etc.
- with  $OH \cdot$ ,  $O_3$ , and  $NO_3 \cdot$  (Yu et al., 2022).

$$173 AOC = \sum_{i} k_{Y_i}[X][Y_i] (2)$$

- where [X] and [Yi] are the number concentrations of molecule oxidant X and Yi,
- respectively, and  $k_{Y_i}$  is the bimolecular rate constant of molecule  $Y_i$  with oxidant  $X_i$ .
- The oxidants included OH, NO<sub>3</sub>, and O<sub>3</sub>(Chapleski et al., 2016). The AOC attributed
- to each reaction rates was extracted during observation box model (OBM) simulations
- using the model's built-in extract rates function. Analyzing the atmospheric chemical
- reactions of typical secondary pollutant O<sub>3</sub> based on the same principle.

Step 2: Tracing and identified key precursor material species influencing AOC by 181 182 examining their roles in photochemical reaction pathways. As a key oxidant and primary driver of AOC, OH· initiates VOC oxidation to produce HO<sub>2</sub>· and 183 RO2 radicals, which subsequently participate in O3 formation and SOA generation 184 185 (Chen et al., 2022; Tadic et al., 2021). Controlling OH, HO<sub>2</sub>, and RO<sub>2</sub> radicals is critical for regulating AOC, particularly through modulating OH· concentrations. The 186 187 study of Yang et al. (2024) demonstrated that alkene-O<sub>3</sub> reactions generate criegee 188 intermediates (CI), which enhance OH, HO<sub>2</sub>, and RO<sub>2</sub> radical concentrations and accelerate RO<sub>x</sub>· cycling (Yang et al., 2024). Elevated RO<sub>2</sub>· and HO<sub>2</sub>· concentrations 189 190 during RO<sub>x</sub> cycling enhances OH production, which is the primary driver of AOC.

- We introduce the radiation equivalent oxidation capacity (REOC) metric based on
- radical generation pathways from intermediate species. REOC quantifies precursor
- contributions to OH·, HO<sub>2</sub>·, and RO<sub>2</sub>· radicals by normalizing their production to
- equivalent OH· oxidation capacity, providing a unified measure of VOCs species
- oxidative impacts. The REOC can be calculated by Eq. (3) (5).

$$REOC = d[OH \cdot]_t + \alpha \times d[HO_2 \cdot]_t + \beta \times d[RO_2 \cdot]_t$$
 (3)

$$\alpha = \frac{\sum_{1}^{n} ([HO_{2} \cdot] \rightarrow [OH \cdot])}{\sum_{1}^{p} P[HO_{2} \cdot]}$$

(4

$$200 \qquad \beta = \frac{\sum_{1}^{r} ([RO_{2} \cdot] \rightarrow [OH \cdot])}{\sum_{1}^{n} P[RO_{2} \cdot]}$$

(5)

210

- Where the  $d[OH\cdot]_t$ ,  $d[OH\cdot]_t$  and  $d[RO_2\cdot]_t$  are the directly generated rates of OH·,
- $HO_2$  and  $RO_2$  radicals at time t. Parameters  $\alpha$  and  $\beta$  represent the conversion
- efficiencies of HO<sub>2</sub>· and RO<sub>2</sub>· to OH·, respectively, which can be calculated through
- dividing the rate of conversion of all HO<sub>2</sub>· and RO<sub>2</sub>· to OH· by the rate of generation
- of all HO<sub>2</sub>· and RO<sub>2</sub>·, respectively. Reaction pathway tracing and analyzing enables
- systematic quantification of OH· radical production from VOCs, more effectively
- characterizing precursor-specific contributions to atmospheric oxidation processes.
- Part 3: Novel framework for source-resolved AOC sensitivity and attribution:
- Integrating PMF with precursor-specific quantification
- Step 1: VOC and NO<sub>x</sub> source apportionments were calculated by the PMF model (US
- EPA 5.0). This study selected thirty-eight InVOCs species and NO<sub>x</sub> for PMF analysis,
- and applies its core principle of decomposing the sampling data matrix into two
- constituent matrices to estimate VOC species contributions (He et al., 2019; Yu et al.,
- 2022; Liu et al., 2025).

$$X_{ij} = \sum_{k=1}^{p} g_{ik} f_{kj} + e_{ij}$$
 (6)

- where  $x_{ij}$  represents the concentration of species j in sample i;  $g_{ik}$  is the contribution of
- source k in the sample i; source profile  $f_{kj}$  is the mass percentage of species j in source
- k;  $e_{ij}$  is the residual for species j in sample i; and p is the total number of source
- categories. For other relevant calculation formulas of the PMF model can be found in
- Text S4.

- Step 2: The sensitivity of AOC to various emission sources was analyzed by
- calculating their Relative Incremental Atmospheric oxidation capacity (RIA). This
- was accomplished by integrating the OBM PMF models to simulate AOC changes
- under various emission reduction scenarios. Through this systematic scenario

- modeling, we quantified source-specific sensitivity coefficients to identify the most
- influential sources. This methodology identifies dominant AOC-controlling emission
- sources through response quantification. The calculation equations of relative
- incremental reactivity (RIR) and RIA are shown in Eq. (7) and Eq. (8), respectively. 234

$$RIR_t = \frac{\text{Net(X)} - \text{Net(X} - \Delta X)/\text{Net(X)}}{\Delta S(X)/S(X)}$$
(7)

$$RIA = \frac{\sum_{1}^{n} RIR_{t} \times AOC}{\sum_{1}^{n} AOC}$$
 (8)

239

- where  $RIR_t$  represents the sensitivity of different emission sources after emission reduction at time t, Net(X) represents the net production rate of a specific species X, Group X, or source X.  $Net(X - \Delta X)$  refers to the net production rate of X caused by the hypothetical emission change  $\Delta X$ . S(X) is the total observed mixing ratio of precursor X.  $\Delta S(X)$  is the total mixing ratio change of precursor X caused by the hypothetical
- emission change (assumed to be 20 % in this study), n is the number of emission 244
- sources derived from PMF. 245

247

249

- Step 3: We further establish a quantification framework assessing both emission source contributions and species-specific impacts on AOC. Integrating PMF source apportionment with relative AOC reactivity metrics, this method systematically determines (1) source-level AOC contributions and (2) within-source VOC species oxidation capacity, identifying dominant emission sources and pollutant species. The
- species and emission source contribution of AOC are shown in Eq. (9) and Eq. (10). 252

$$SCAOC_{ij} = \frac{RIA \times kOH_{ij}}{\sum_{1}^{n} RIA \times kOH_{ij}}$$
(9)

$$255 CAOC_i = \frac{RIA \times AOC_i}{\sum_{1}^{n} RIA \times AOC_i} (10)$$

- Where  $SCAOC_{ij}$  is the contribution of species j in source i to AOC,  $kOH_{ij}$  is the reaction rate constant between VOCs species and OH· radicals of species i in source i, 258 259 which used to characterize the contribution of VOCs species to the chain reaction of
- free radicals, CAOC<sub>i</sub> is the contribution of source i to AOC, AOCi is the AOC of 260

source i derived from OBM-PMF, n is the number of emission sources derived from

PMF. 262

264

268

# Part 4: The regulation results of AOC

Fig. 1. shows the workflow of the AOCPT method. Briefly, (1) the AOC of each time steps during the study period was quantified by OBM, and identified the reactions and oxidants that contribute significant to AOC. (2) Through pathway tracing and analyzing of atmospheric chemical reactions, we developed the REOC metric to systematically quantify VOCs-driven OH· radical production, identifying key reactive VOCs species. (3) The PMF-based source apportionment identifies emission source

sensitivities influencing AOC, while quantitatively assessing source-specific contributions from individual VOC species and NO<sub>x</sub> to AOC variations, and analyzed the contributions of different emission sources to AOC. Overall, achieving path tracing and traceability of AOC. Compared to existing studies, the AOCPT method proposed in this study conducts quantitative and qualitative analysis from the perspective of secondary pollutant formation, rather than focusing solely on individual secondary pollutant. It provides a methodological basis and research direction for the synergistic control and management of secondary pollutants.

Fig. 1. The workflow of the AOCPT method.

#### 3 Results and discussion

## 3.1 Overview characteristics

Studies have indicated that the concentration of measured volatile organic compounds (MVOCs) was lower than Initial VOCs (InVOCs) (Wang et al., 2022b). Therefore, the InVOCs have been analyzed in this study (Text S5 and Fig. S1.). During the daytime (8:00 to 18:00), the average concentration of InVOCs ( $20.1 \pm 1.0$  ppbv) was higher than MVOCs ( $15.3 \pm 2.6$  ppbv) 30.0%. Especially, undervalued the concentration of alkene (Isoprene and anthropogenic alkene was undervalued 34.8% and 29.9%, respectively), which play an important role in photochemical reaction processes (Yang et al., 2024). This difference was defined as consumed VOCs, which the VOCs consumed to participated in atmospheric photochemical reactions (Wang et al., 2022b).

The average concentrations and diurnal variation characteristics of atmospheric

302

304

312313

314315

317318

321

pollutants (O<sub>3</sub>, InVOCs, CO and NO<sub>2</sub>) from study period were analyzed (Fig. 2.). O<sub>3</sub> is a typical representative of secondary pollutants in summer. During pollution period (O<sub>3</sub> > 160 μg/m<sup>3</sup>), the average concentration of NO<sub>2</sub>, CO, O<sub>3</sub> and InVOCs was higher than clean period ( $O_3 \le 160 \mu \text{g/m}^3$ ) 21.9%, 21.7%, 22.9% and 77.2%, respectively. The increase in concentration of oxidants (NO2, CO and O3 etc.), which can helps to enhance the AOC capability (Liu et al., 2021). The CO and NO<sub>2</sub> showed unimodal variation characteristics (the highest in 8:00), and the concentration of pollution period were higher than clean period during 8:00 to 12:00 46.7% and 119.6%, respectively. However, the InVOCs showed bimodal variation characteristics (the highest in 8:00 and 14:00), and the concentration of pollution period were higher than clean period during 8:00 to 12:00 49.7% and 89.8%, respectively. This shown that the precursors were accumulation in the morning and increased in daytime, which may promote strong photochemical reactions, especially in the afternoon (12:00 to 16:00), promote the enhancement of AOC capability and leading to O<sub>3</sub> pollution (Liu et al., 2022). The highest d-value of InVOCs and MVOCs was in 14:00 (50.1%), which also indicated the strong photochemical reactions in afternoon (Fig. S1.). Especially Isoprene and anthropogenic alkene between InVOCs and MVOCs, which d-value were largest, due to the strong photochemical reactions during 12:00 to 16:00. Diurnal variation patterns demonstrate that enhanced precursor emissions coupled with chemical depletion drive summer secondary pollution events, which substantiating the implementation basis for the secondary pollution control methods in this study.

Fig. 2. Diurnal variations in concentrations of atmospheric pollutants during polluted period and clean period ((a), (b), (c) and (d) were CO, O<sub>3</sub>, NO<sub>2</sub> and InVOCs, respectively)

#### 3.2 Species tracing and analyzing of atmospheric oxidizing capacity

#### 3.2.1 Quantification of atmospheric oxidizing capacity

The AOC during the sampling periods was quantified, as shown in Fig. 3. The 326 calculated averaged value of total AOC was 5.5×107molec·cm<sup>-3</sup>·s<sup>-1</sup>, with a pollution 327 period AOC of 7.7×10<sup>7</sup>molec·cm<sup>-3</sup>·s<sup>-1</sup>, which was 126.3% higher than the clean 328 period (3.4×10<sup>7</sup>molec·cm<sup>-3</sup>·s<sup>-1</sup>). The AOC from pollution period was higher than 329 Zhengzhou (6.2×10<sup>7</sup>molec·cm<sup>-3</sup>·s<sup>-1</sup>in 2020) (Yu et al., 2022), Shanghai (approx. 330 331  $3.7 \times 10^7 \text{molec} \cdot \text{cm}^{-3} \cdot \text{s}^{-1}$ (Zhu et al., 2020) and Hongkong(approx. 6.78×10<sup>7</sup>molec·cm<sup>-3</sup>·s<sup>-1</sup>) (Xue et al., 2016). Higher AOC serves as an important driver 332 of secondary pollution incidents in summertime (Zhu et al., 2020). Meanwhile, this 333 establishes favorable operational parameters for AOC investigations within the study 334 framework. During pollution period, OH exhibited the highest average concentration 335 336 (3.8×10<sup>7</sup> molec·cm<sup>-3</sup>·s<sup>-1</sup>) in AOC, followed by O<sub>3</sub> (2.8×10<sup>7</sup> molec·cm<sup>-3</sup>·s<sup>-1</sup>) and  $NO_3$ ·  $(1.2\times10^7 \text{ molec·cm}^{-3}\cdot\text{s}^{-1})$ , contributing 48.7%, 35.7%, and 15.5%, respectively. 337 Thus, OH· was the main contributor of atmospheric oxidation, aligning with findings 338 339 from other studies in diverse geographical regions (Yu et al., 2022; Guo et al., 2022; Zhang et al., 2021). 340

341342

343344

352

355

362363

Our further mechanistic analysis of AOC associated reactions elucidates summertime secondary pollution formation (Fig. 3). The average contribution of O<sub>3</sub> + NO<sub>2</sub> reactions to AOC during pollution period (36.2%) exceeds that during clean period (25.9%), particularly between 8:00 to 12:00, where it exceeded clean periods by an average of 20.7%. Elevated ambient NO<sub>2</sub> concentrations (Fig. 2c) combined with attenuated O<sub>3</sub> titration establish critical preconditions for this reaction mechanism (Dong et al., 2023). The  $O_3 + NO_2$  promotes  $O_2$  generation, facilitating  $RO + O_2$  to HO<sub>2</sub>·, which enhances the production of OH· radical from HO<sub>2</sub>· + NO reaction and exacerbates the AOC (Wang et al., 2017). Diurnal NO<sub>2</sub> decline and VOCs accumulation (Fig. 2), coupled with enhances photochemical activity driving intensified the OH + VOCs reactions. Especially polluted periods exhibit an 18.3% higher daytime average in OH· + VOCs reactions compared to clean periods, which directly supports the reactions of RO<sub>2</sub>· + NO. That's also why the maximum reaction rates of HO<sub>2</sub>· + NO and RO<sub>2</sub>· + NO during the pollution period were 85.5% and 113.9% higher than those during the clean period, respectively (Fig. S2.). During the cleaning period, VOCs emissions are more prominent than NO<sub>x</sub> emissions (Fig. 2), make the daytime OH· + VOCs dominate OH· reactions contributions of AOC during clean period (37.9%). Overall, O<sub>3</sub> + NO<sub>2</sub> and OH· + VOCs were the mainly reaction of AOC, which collectively accounted for 48.5 to 56.1% of daytime AOC during the sampling period. Therefore, controlling NO<sub>2</sub> and InVOCs emissions were essential to mitigated AOC and secondary pollution incidents in summer. However, the emission sources and species of InVOCs were complex. Thus, it is important to tracking and identifying key VOCs that have a significant impact on AOC through free radical chemistry.

367368369

373374

379380

385

388

391

Fig. 3. Diurnal patterns of AOC simulated

# 3.2.2 Free radical budget analysis

The free radicals during different pollution periods in the study period were analyzed through the F0AM model (Fig. 4.). The OH and HO<sub>2</sub> showed unimodal variation characteristics during the pollution period, average concentration were 3.6×10<sup>6</sup> molecules cm<sup>-3</sup> and 0.4×10<sup>9</sup> molecules cm<sup>-3</sup>, which higher than clean period 62.3% and 38.6%, respectively. During the pollution period, the maximum of OH· in this study (13.0 ×106 molecules·cm<sup>-3</sup>) was higher than Shanghai (approx. 9.5×106 molecules · cm<sup>-3</sup>) (Zhang et al., 2021), Lanzhou (4.5×10<sup>6</sup> molecules · cm<sup>-3</sup>) (Guo et al., 2022), and Beijing (2.7×10<sup>6</sup> molecules·cm<sup>-3</sup>) (Slater et al., 2020), and the maximum of HO<sub>2</sub>· (1.31×10<sup>9</sup> molecules·cm<sup>-3</sup>) was higher than Beijing (7.3×10<sup>8</sup> molecules·cm<sup>-3</sup>) (Jia et al., 2023) and Shanghai (approx. 3.77×10<sup>8</sup> molecules·cm<sup>-3</sup>) (Zhu et al., 2020). The OH constitute the predominant regulator of atmospheric oxidation processes (Yu et al., 2022), governing the initiation and propagation of radical chain reactions in the troposphere (George et al., 2023). Meanwhile, OH contributed to the decomposition of precursor VOCs, which was important to the secondary pollution incidents in summer. Moreover, the reaction of HO<sub>2</sub>· + NO can further promote the generation of OH radicals. The higher free radicals concentrations in this study indicated higher atmospheric oxidation, which the linear relationships between AOC and OH· radicals with a fitting degree of R<sup>2</sup>=0.77 (Text S7). Thus, the reaction pathways of OH· radicals in photochemical processes were employed to trace critical VOCs and primary emission sources, which enabled the regulation of AOC and thereby subsequent reduction of secondary pollution, establishing this approach as a viable control strategy.

398399

406

410

413

416

Fig. 4. Diurnal variations of free radical during the polluted and clean period.

## 3.2.3 source of free radical

During the observation period, the radicals cycling process in the daytime (8:00 to 18:00) was shown in Fig. 5. OH· plays a vital role in the  $RO_x$ · (OH· + HO<sub>2</sub>· + RO· + RO<sub>2</sub>·) cycle in photochemical reactions through InVOCs to the secondary pollution formation in summer (Wei et al., 2023; George et al., 2023; Yang et al., 2024). The OH was mainly producted by HO<sub>2</sub> + NO, the reaction rate of pollution period was  $8.6 \pm 5.8$  ppbv/h<sup>-1</sup> higher than clean period 32.3%, which was also the dominated reaction of the secondary pollution formation during summertime (as shown in 3.2.1). Subsequently, OH· + InVOCs to generated RO<sub>2</sub>·, which reaction rate was  $5.3 \pm 3.6$ ppbv·h<sup>-1</sup> in pollution period, higher than clean period (3.9  $\pm$  1.9). OH· + alkene was the dominated reaction, which accounted for 50.9% during the pollution period.  $RO_2$  + NO to generated RO, the rate during pollution period (6.7 ± 5.0 ppbv·h<sup>-1</sup>) was 52.3% higher than the clean period  $(4.4 \pm 2.1 \text{ ppbv} \cdot \text{h}^{-1})$ , which was another dominated reaction the secondary pollution formation in summertime (as also shown in 3.2.1). Meanwhile, RO + O<sub>2</sub> to generated HO<sub>2</sub>: (reaction rate was  $8.9 \pm 6.0$  ppbv·h<sup>-1</sup>), which increased rapidly the HO<sub>2</sub>. Noteworthily that alkene can directly reacted with O<sub>3</sub> to productid criegee intermediates (CI), which can increase the concentrations of RO<sub>2</sub>. OH·, and HO<sub>2</sub>· radicals (Yang et al., 2024). Therefore, the reaction of alkene + O<sub>3</sub> and OVOCs + hv can be considered as the direct source of OH·, RO<sub>2</sub>· and HO<sub>2</sub>· radicals, which produced from primary pollutant. To mitigate the radical reaction processes, it is essential to regulating their emission sources and mainly species.

Fig. 5. The average daytime (8:00 to 18:00) budget of the ROx radical cycle, with reaction rates shown in ppbv·h<sup>-1</sup>. Primary radical sources and sinks are highlighted in yellow and pink. Blue arrows denote ROx recycling pathways. Reaction rates for polluted and clean periods are displayed in orange and green text, respectively.

423 424

426

432

435

437

440

442443

445446

448

421

> This study employed the REOC concept (Eq. (3)), which was used to unify quantification the contribution of InVOCs to radical generation (Fig. 6.). Due to the predominance of OH:-related reactions in AOC (as also shown in 3.2.1), we used REOC to normalizes the ability of InVOCs to generate different radicals as the ability to generate OH· radicals, which indirectly reflecting the contribution of InVOCs to AOC. The reaction of alkene + O<sub>3</sub> influenced the concentrations of RO<sub>2</sub>, OH· and HO<sub>2</sub>· contributing 93.4%, 73.9% and 58.0%, respectively. Trans-2-butene was identified as a key source species, contributing 76.3% and 60.3% to the formation of RO2, and OH, respectively. Previous studies have demonstrated that trans/cis-2-butene and pentenes readily react with O<sub>3</sub>, generating CH3CHOOB criegee intermediates, which rapidly decompose into CH3O2, OH·, and CO (Yang et al., 2024). This process propagates the RO<sub>x</sub> cycle, especially the OH and CO are both key oxidants in the AOC reaction (Fig. 3), which ultimately drives significant AOC and secondary pollution formation in summertime. Therefore, to better assess direct InVOCs contributions to AOC, we developed the REOC metric, which quantifies radical-mediated oxidative impacts by normalizing VOC-derived RO<sub>2</sub> and HO<sub>2</sub>· production to OH·-equivalent values through chemical reaction pathways. This framework identifies localized InVOCs species critically influencing AOC, with trans-2-butene demonstrating predominant REOC contributions (71.1%) followed by trans-2-pentene (12.5%). Although the species identified by the method of REOC may have a relatively small proportion in TVOC, but high reactivity allows it to have a significant impact on atmospheric photochemical pollution even at lower concentrations (Yang et al., 2020). Thus, precursor emission control strategies must prioritize emission sources, that release key components and species demonstrating considerable impacts on AOC, rather than focusing solely on total emission reduction targets. The methodology of REOC establishes a reactivity-based prioritization system for targeted precursor species management.

455 456

463

465

471

472473

476

Fig. 6. Daytime (8:00 to 18:00) average contributions of initial sources to  $OH^{\cdot}$ ,  $HO_{2^{\cdot}}$ , and  $RO_{2^{\cdot}}$  during the observation period

#### 3.3 Source apportionment and emission reduction

#### 3.3.1 Secondary pollutant precursors source apportionment

This study applied the PMF 5.0 model to analyze the secondary pollutant precursors sources (Fig. 7.). During the sampling period (Fig. 7. (a)), diesel vehicles emission (26.3%), gasoline vehicles emission (25.3%), and industrial process (18.0%) dominated InVOCs sources. Especially during pollution episodes (Fig. 7. (b)), the contribution of diesel vehicles emission (30.7%) was dominated to InVOCs, followed by industrial process (20.6%), and gasoline vehicles emission (23.8%). Notably, diesel vehicles emission and industrial process exhibited 11.5% and 6.8% higher InVOCs contributions during pollution periods than in clean periods, respectively. NO<sub>x</sub> primarily originated from diesel vehicles emission (30.2%), gasoline vehicles emission (29.1%), industrial process (20.6%), and combustion source (20.2%) (Fig. 7. (d)). Contributions from diesel vehicles emission and industrial process to NO<sub>x</sub> during pollution period exceeded clean periods by 11.2% and 6.3%, respectively. Collectively, diesel vehicles emission and industrial process contributed more to both InVOCs and NO<sub>x</sub>, particularly during pollution period, likely driven by industrial expansion and heightened transport demands. In 2024, the mining industry (accounting for 76.2% of the industrial total) registering a 6.3% growth in Changzhi City, which the location of the research case (Czmbs, 2024). Coupled with an energy mix heavily reliant on thermal power (91.8% vs. 8.5% from renewables) intensified emission pressures (Czmbs, 2024). Thus, to mitigate the precursors of secondary pollution, industrial

cities should prioritize emission controls for heavy industries.

Fig. 7. Source contribution of secondary pollutant O<sub>3</sub> precursor from the PMF model. a, b and c were InVOCs. d, e and f were NO<sub>x</sub>

# 3.3.2 Species and source apportionment of AOC

Based on the source apportionment results (section 3.3.1), critical sources affecting the AOC were identified (Fig. 8.), and the contributions of key InVOCs species from these sources were analyzed (Fig. 9.). During the sampling period, industrial process (26.8%) and diesel vehicle emission (24.1%) were the dominant contributors to AOC, followed by solvent utilization (17.9%), combustion source (13.4%), biogenic source (10.3%) and gasoline vehicle emission (7.6%). Especially the industrial process (33.0%) and diesel vehicle emission (28.8%) during polluted periods demonstrate 17.8% and 13.5% elevation compared to cleaning period, respectively.

493 494

498

482

484

487

490

Among VOCs species contributions across emission sources, we prioritized alkenes, which demonstrating significant impacts to AOC (section 3.2.3). Industrial process exhibited the highest alkene contributions (31.0%), followed by diesel vehicles emission (20.7%). Source-specific alkene contributions were significantly correlated (P 

processes. Especially, with the analysis of the key alkene species trans-2-butene (section 3.2.3), which disproportionately affects AOC, revealed its highest impact from industrial process (49.3%), followed by diesel vehicle emission (20.6%). Trans-2-butene emission magnitudes across sources exhibited significantly correlations (P < 0.05,  $R^2 \approx 0.91$ ) with their corresponding AOC contributions.

However, gasoline vehicle emission exhibited 41.1% higher total VOCs emissions than industrial process, primarily attributed to elevated contributions from ethane, propane, isopentane, and ethylene. But for trans-2-butene, which has a higher impact on AOC, gasoline vehicle emission exhibited 83.5% and 65.8% lower emissions compared to industrial process and diesel vehicle emission, respectively. While previous studies have shown that high emission levels may offset low chemical reactivity of VOCs species (Tang et al., 2018), the case of this study demonstrates that high-reactivity species remain critical concerns, particularly regarding their impacts on AOC. This also indicated that if the current secondary pollution control strategies focusing solely on high VOCs emission sources and neglecting the impact of source emissions on AOC, particularly for sources with lower aggregate emissions but elevated reactive species emissions, it may lead to survivorship bias in the implementation effectiveness of control measures. This discrepancy may underlie persistent summertime secondary pollution episodes despite substantial precursor reductions.

For  $NO_x$ , the predominantly influence AOC originate from diesel vehicle emission (52.0%) and industrial process (28.5%), followed by combustion source (9.8%) and gasoline vehicle emission (9.7%). A statistically significant correlation (P < 0.05) exists between source-specific  $NO_x$  contributions and their AOC impacts. This may be attributed to elevated  $NO_x$  emissions enhancing  $O_3 + NO_2$  reactions, particularly during morning period (section 3.2.1), thereby increasing the source contributions to AOC. Therefore, necessitating integrated control strategies targeting both VOC and  $NO_x$  emission sources for effective AOC mitigation.

A comparison between AOC and O<sub>3</sub> source apportionment was conducted using summertime O<sub>3</sub> pollution of the case study (Fig. S5). The analysis of O<sub>3</sub> source apportionment, which identified industrial emission (22.6%), gasoline vehicle emission (22.1%), and combustion source (21.3%) as primary contributors, systematically underestimated diesel vehicle emission (8.3% underestimation) and industrial emission (4.2% underestimation) source impacts while overestimating others. The differences in source apportionment results may directly affect the direction of pollution emission control. Thus, compared to O<sub>3</sub> source apportionment approaches, AOC oriented source tracing may better facilitate coordinated secondary pollution control, through its comprehensive consideration of the conversion process from primary pollutants to secondary pollutants.

Fig. 8. Source contribution of key primary pollutants from AOC source apportionment.

Fig. 9. The contribution of various species in the emission source to AOC.

# 3.3.3 Analysis of emission reduction sensitivity

This study further analyzes source sensitivities of AOC, O<sub>3</sub>, and secondary organic aerosols (SOA) to precursors (Fig. 10.). Given that the self-reaction rate between peroxy radicals (self-reaction) is typically used to characterize the formation potential of secondary organic aerosols (SOA) (Lyu et al., 2022), we used it as a marker to evaluate the generation of secondary pollutants (detailed calculation method of self-reaction showed in Text S9).

570571

577578

584

588 589

591

AOC demonstrates the highest source sensitivity to industrial process (0.041), followed by diesel vehicle emission (0.037) and solvent utilization (0.029). Compared to AOC source sensitivities (Fig. 10. a), O<sub>3</sub> sensitivity analysis (Fig. 10. b) exhibits 28.7%, 26.5%, and 48.5% underestimation for industrial process (0.029), solvent utilization (0.021), and diesel vehicle emission (0.019), respectively, while overestimating gasoline vehicle emission (0.018) and combustion sources (0.024) by 48.8% and 14.4%. Similarly, self-reaction sensitivity analysis (Fig. 10. c) shows 25.7%, 13.4%, and 5.6% underestimation for industrial process (0.030), solvent utilization (0.025), and diesel vehicle emission (0.035) compared to AOC, contrasted by 172% and 25% overestimation for gasoline vehicles (0.033) and combustion sources (0.026). Previous studies have identified industrial process and combustion sources have a significant impact on O<sub>3</sub> pollution, primarily due to their elevated precursor pollutants emissions that in promoting O<sub>3</sub> formation (Zhan et al., 2023). Additional research has also established industrial process and vehicular emissions of semivolatile and intermediate-volatility organic compounds (SVOCs and IVOCs) as dominant precursors in SOA formation (Tang et al., 2021; Miao et al., 2021). However, these studies remain confined to single secondary pollutant analyses, neglecting the control of secondary pollution from the perspective of AOC, especially the lack of analysis of alkenes like trans-2-butene etc., which crucially AOC. Thus, given that AOC quantifies secondary pollutant formation potential (Yu et al., 2022), the source sensitivity divergence with both AOC and individual secondary pollutants (e.g., O<sub>3</sub> and SOA) indicates that it was necessitates prioritizing emission sources' oxidation capacity impacts over their singular pollutant contributions (Wang et al., 2024).

Fig. 10. source sensitivity analysis. (a), (b) and (c) represents the sensitivity of AOC, O<sub>3</sub>, and SOA to different emission sources, respectively.

## 3.3.4 impact of reduction scenarios on secondary pollutant generation

The isopleth diagram was used in this study to quantify the nonlinear relationship between AOC, O<sub>3</sub> and SOA(Fig. 11.) with the precursors (InVOCs and NO<sub>x</sub>), by using the F0AM-MCM model (Niu et al., 2024; Mozaffar et al., 2021). Initially, the average daytime concentrations of NO<sub>x</sub> and VOCs are used as baseline. Subsequently,

https://doi.org/10.5194/egusphere-2025-4355 Preprint. Discussion started: 17 October 2025 © Author(s) 2025. CC BY 4.0 License.

VOCs and NO<sub>x</sub> are varied at 10 % intervals, respectively, and a total of 441 analysis scenarios were constructed. Subsequently, VOCs and NO<sub>x</sub> are varied from -60% to 90% at 10 % intervals, respectively, to construct the scenario matrix.

597598

602

612

617

623

626

As shown in Fig. 11, the isopleth analysis indicates that reductions in both VOCs and NO<sub>x</sub> lead to a decrease in the AOC, net O<sub>3</sub> production rate (Net O<sub>3</sub>), and self-reaction in the studied city. Notably, NO<sub>x</sub> reduction has a more pronounced effect on the decrease in AOC. This may be associated with the high contributions of OH· + VOCs and  $O_3 + NO_2$  in the specific reaction of AOC in the case (as shown in 3.2.1). Firstly, OH· + VOCs generates substantial RO<sub>2</sub>· radicals, and NO acts as a catalyst to accelerate the regeneration of OH· radicals from RO<sub>2</sub>· in the RO<sub>x</sub>· cycle, while AOC is largely determined by OH· radicals. Secondly, NO<sub>2</sub> directly promotes the O<sub>3</sub> + NO<sub>2</sub> reaction. We established the reduction targets based on the average levels during the cleaning period for AOC (5.8×10<sup>7</sup> molec·cm<sup>-3</sup>·s<sup>-1</sup>), Net (O<sub>3</sub>) (9.3 ppbv·h<sup>-1</sup>), and self-reaction (1.1 ppbv·h-1), respectively. To achieve the AOC target, the VOCs reduction of at least 60% was required if NO<sub>x</sub> emissions were unchanged, whereas the 40% NO<sub>x</sub> reduction was necessary if VOCs emissions remain constant. However, achieving independent reductions is challenging due to the similarities in the sources of VOCs and NO<sub>x</sub> emission. Therefore, to meet the target for AOC, a simultaneous reduction of 60% in VOCs and 30% in NOx was required (Fig. 11, b). Meanwhile, to meet the target for the Net (O<sub>3</sub>), a coordinated reduction of at least 20% in VOCs and 10% in NO<sub>x</sub> was needed (Fig. 11, d). Previous research has shown that the co-reduction of VOCs and NO<sub>x</sub> is critical for controlling O<sub>3</sub> pollution. Specifically, the reduction strategy targeting AOC results in a more pronounced decrease in the Net (O<sub>3</sub>), as indicated by the red arrow from point (b) to (d) in Fig. 11. In contrast, a reduction strategy designed solely for O<sub>3</sub> was not sufficient to meet the reduction target for self-reaction, by the blue arrow from point (d) to (f) in Fig. 11. A notable complication is the observed negative correlation between self-reaction and NO<sub>x</sub>. This implies that a substantial reduction in NO<sub>x</sub> could counter-intuitively cause self-reaction to increase, which could be counterproductive for SOA control. Despite this, when we assess self-reaction using the AOC-based reduction scenario (at least 60% for VOCs and 30% for NOx), it fully satisfies the reduction target for self-reaction (as indicated by the red arrow from point (b) to (f) in Fig. 11). This result provides compelling evidence that a reduction strategy based on AOC enables the simultaneous mitigation of both O<sub>3</sub> and SOA. Therefore, an AOC-centric approach offers a viable pathway for the synergistic control of secondary pollutants.

Fig. 11. Response of AOC,  $Net(O_3)$  and self-reaction to different VOCs and  $NO_x$  reduction percentages derived from the empirical kinetic modeling approach. Red dots from (a), (c), and (e) represent the baseline scenario (average levels without precursor pollutant controls during the study period). Black lines from (b), (d), and (f) indicate target levels to be achieved by precursor control schemes (average during cleaning periods). Red arrows show the effects of the AOC reduction scheme on achieving  $O_3$  and SOA targets. Blue arrows show the effects of the  $O_3$  reduction scheme on achieving the SOA target.

#### **4 Conclusion**

This study developed and applied the atmospheric oxidation capacity formation path tracing (AOCPT) approach, a framework for guiding secondary pollutant control. This method employs the developed radiation equivalent oxidation capacity (REOC) metric to systematically quantify VOCs driven OH radical production, which indirectly enables the standardized quantification of key precursor species influencing

AOC. The defined relative incremental atmospheric oxidation capacity (RIA) method directly quantifies the impact of emission sources on AOC. Furthermore, it further quantifies the contributions of different precursor species and emission sources to AOC, which using a refined AOC source analysis method. This AOCPT approach offers new insights for secondary pollution control from the perspective of AOC.

651652653

655

668

674

A field application of this methodology revealed that OH related reactions were the dominant driver of AOC (56.9%), and daytime contributions from O<sub>3</sub> + NO<sub>2</sub> and OH· + VOCs reactions being particularly prominent (48.5–56.1%). This underscores the necessity of co-reducing both VOCs and NOx for effective AOC regulation. The REOC analysis identified trans-2-butene as a critical contributor to AOC (71.1%). Consequently, further analysis pinpointed industrial processes (26.8%) and diesel vehicle emissions (24.1%) as the primary AOC sources, largely attributed to their emissions of trans-2-butene (accounting for 49.3% and 20.6% of total trans-2-butene, respectively). These findings provide direct, quantifiable evidence linking specific VOCs species and emission sources to the overall AOC, offering clear and actionable targets for regulatory action. Critically, conventional sensitivity analyses based on ozone (O<sub>3</sub>) and self-reaction were found to significant underestimate the contributions from industrial processes (by 28.7% and 25.7%, respectively) and diesel vehicles (by 48.5% and 13.4%, respectively) compared to our AOC-based assessment. This discrepancy can introduce substantial bias into policymaking. Crucially, our scenario analysis reveals that O<sub>3</sub>-targeted abatement can inadvertently increase secondary organic aerosols (SOA) levels, leading to a skewed mitigation outcome akin to "survivor bias". In contrast, an AOC-centric strategy achieves significant and simultaneous reductions in both O<sub>3</sub> and SOA. This provides definitive evidence that compared to traditional treatment of single secondary pollutants, pollution abatement strategy based on AOC regulation can achieve refined co-mitigation of secondary pollutants. Therefore, the AOC-based approach for secondary pollution control serves not merely an alternative but also enhances the comprehensiveness and effectiveness of control strategies to some extent.

676677678

As China confronts a plateau in air quality improvement, where significant reductions in primary pollutants have not yielded proportional decreases in secondary pollution, a new strategy is urgently needed. This study argues that breaking the current bottleneck requires a fundamental shift in perspective. This paradigm shift, from pollutant-specific control to regulating the atmosphere's overall oxidative capacity, represents a pivotal step forward, offering a scientifically robust path to overcome the current impasse and achieve sustainable, long-term air quality goals.

685 686

## **Author contributions**

YK: Writing - original draft, Methodology, Investigation, Data curation. YY: Writing
 review & editing, Validation, Supervision, Project administration, Methodology,
 Conceptualization. YN: Validation, Supervision, Investigation, Data curation. CY:
 Data curation, Project administration, Conceptualization. JD & YZ & DW:

- Investigation, Data curation. JL & ZL: Validation, Methodology. LP: Validation,
- Supervision, Project administration, Conceptualization.

#### Acknowledgements

- This work was supported by the National Natural Science Foundation of China
- (NSFC) (Grant, No. 42330606, 42422508, 42273058, 22106044), Fundamental
- Research Funds for Central Universities (2024XKRC058, 2024JBZY016) and Shanxi
- Provincial Basic Research Program (Free Exploration Category) Youth scientific
- research project (Grant, No. 202303021212370).

701

703704

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
