# Peer review of "An Observation-Based Methodology and Application for Future"

_EGUsphere, 2025_

## Author Comment (AC1)

**Detailed Response to Reviewers' comments**
**Manuscript Number:** egusphere-2025-4355
**Manuscript Title:** An Observation-Based Methodology and Application for Future Atmosphere Secondary Pollution Control via an Atmospheric Oxidation Capacity Path Tracing Approach
**Note:** The Changes of reviewer's suggestion in the revision manuscript were indicated by the red font.

**Reviewer #1:** The manuscript titled "An Observation-Based Methodology and Application for Future Atmospheric Secondary Pollution Control via an Atmospheric Oxidation Capacity Path Tracing Approach" establishes a framework that can analyze the dominant precursors and sources of AOC. The results provide improved technical pathways for mitigating secondary pollution. I recommend acceptance after the following issues are addressed.

**Response:**

Thank you for your review. For the comments from reviewers, we have revised the manuscript as better as we can.

1. Abstract: (1) Some sentences are too complex, such as "This approach quantitatively traces AOC to its precursors and sources, thereby facilitating the coordinated control of secondary pollution, by integrating three modules … to assess source impacts, and a modified source apportionment technique to resolve the respective contributions of both precursor species and sources to AOC." in lines 26-34. Please simplify corresponding texts or cut them into several short sentences. (2) The AOCPT approach was applied in Changzhi rather than validated; therefore, the description "successfully validated in a field study in Changzhi" in line 34 should be revised. (3) the description of the application results is insufficient. Please provide additional relevant details.

**Response:**

Thank you for your review. We have revised the Abstract following your suggestions.

(1) We have broken down the complex sentence in lines 26-34 into shorter sentences to improve readability. The revisions are as follows (Line 26-33):
"In this study, the Atmospheric Oxidation Capacity Path Tracing (AOCPT) approach was proposed to quantitatively trace AOC to its precursors and sources. It facilitates coordinated control by integrating three core modules. It employs a Radiation Equivalent Oxidation Capacity (REOC) method to quantify precursor species contributions. Meanwhile, it utilizes a Relative Incremental AOC (RIA) metric derived from a coupled box-receptor model to assess source impacts. Finally, a modified source apportionment technique was applied to resolve the respective contributions of both

precursor species and sources to AOC."

(2) We have improved the accuracy of the wording. Specifically, "validated" in line 34 has been changed to "applied".

(3) We have enriched the description of the application results in the abstract as suggested. Specifically, we incorporated key quantitative findings regarding source underestimation. Given the word limit, we focused on adding the most significant quantitative conclusions to ensure the abstract remains concise yet informative. The revisions are as follows (Line 36-40):
"Notably, conventional sensitivity analyses based on ozone ($O_3$)-targeted strategies were found to underestimate the contributions of these two sources by 28.7% and 48.5%, respectively. Furthermore, while $O_3$-targeted abatement inadvertently enhanced secondary organic aerosols (SOA), an AOC-centric strategy enabled the co-mitigation of both pollutants."

2. The second and third paragraphs in the Introduction section should be simplified, and the structure should be further adjusted to make it more logical. And add the applied details in the last paragraph.

**Response:**

We sincerely appreciate your constructive suggestion. We have simplified the text and reorganized the logical structure of the Introduction as suggested. Additionally, details regarding the field application have been incorporated into the last paragraph.

The revised second paragraph is presented as follows:
"In recent years, the Chinese government has achieved substantial reductions in primary pollutant emissions. With the implementation of "the Air Pollution Prevention and Control Action Plan" and "The Three-Year Action Plan on Defending the Blue Sky", the NOx and CO, etc. concentrations of China were reduced by 31% and 33.3%, respectively, in 2024 compared to 2018 (Mep, 2024). However, the environmental risks stemming from secondary pollutants remain a major concern. In 2024, $O_3$-8H concentration in key economic clusters like the Beijing-Tianjin-Hebei (187 $\mu g/m^3$) and the Fenwei Plain region (187 $\mu g/m^3$) remained high (Mep, 2024). Similarly, while primary organic aerosols (POA) have decreased significantly, the reduction in secondary organic aerosols (SOA) has been limited (Chen et al., 2024). This persistence of secondary pollution is directly correlated with strong AOC (Huang et al., 2014; Wang et al., 2022a; Hu et al., 2017). AOC not only drives the photochemical formation of $O_3$ but also contributes of up to 80% to SOA formation (Huang et al., 2014; Zhao et al., 2020). Consequently, the health burdens caused by secondary pollutants, such as respiratory and cardiovascular diseases, continue to rise (Zhang et al., 2022) These trends indicate that current precursor emission control policies have failed to effectively mitigate the impacts of secondary pollution."

The revised third paragraph is presented as follows:

"Previous studies have indicated that refined emission control strategies are more effective than broad reduction policies. For instance, source-specific mitigation measures based on speciated emission inventories have been proposed to control $O_3$ pollution (Wu and Xie, 2017). Others have highlighted the need for simultaneous control of VOCs and NOx (Ding et al., 2022) or identified critical VOC species based on $RO_2$ radical chemistry (Liang et al., 2024). However, most of these studies neglected the importance of AOC, which acts as the principal driver of secondary pollution generation (Li et al., 2024). Even among existing studies of AOC, the focus predominately remains on chemical mechanisms and radical interactions (Yu et al., 2022; Mochida et al., 2003), with limited exploration of emission-driven oxidation dynamics. This gap is critical because attempts to control $O_3$ alone can unexpectedly increase SOA levels in regions with high AOC due to complex, non-linear mechanisms (Niu et al., 2024a; Lyu et al., 2022). Focusing solely on individual secondary pollutants creates a "survivor bias" and may result in deviations in emission reduction strategies (Le et al., 2020; Galbally, 2007). Therefore, it is essential to prioritize the investigation of AOC and identify source contributions to AOC for the coordinated control of secondary pollution. Consequently, current strategies relying on source analysis for individual secondary pollutants have limitations. In contrast, direct traceability analysis of AOC offers a more representative perspective and enhances regulatory efficacy for secondary pollution control."

The revised last paragraph is presented as follows:

[revised manuscript text omitted]

3. Introduce the QA/QC about sample and analysis.

**Response:**

Thank you for your review. We have added the detailed QA/QC procedures for both sampling and analysis in the supplementary material Text S2 as suggested. Accordingly, the citation in the main text has been revised to (Line126-128):

"The detailed sampling procedures, analytical protocols, and strict quality assurance and quality control (QA/QC) measures are presented in Text S2."

Specifically, we have supplemented Text S2 with information regarding:

"Target compounds were identified based on their retention times and mass spectra, and quantified using a multi-point external calibration method. The calibration standards were prepared by dynamically diluting a 100 ppbv PAMS (Photochemical Assessment Monitoring System, Spectra, USA) standard gas mixture to concentrations of 0.5, 1, 5, 15, 50, and 100 ppbv.

Strict QA/QC procedures were implemented throughout the sampling and analysis processes. Before sampling, all SUMMA canisters were cleaned 3 to 5 times using a canister cleaning system. After cleaning, 10% of the canisters were selected for blank tests by filling them with high-purity nitrogen to ensure the background concentrations were below the method detection limits (MDL). To prevent experimental errors, laboratory blank samples were analyzed prior to each batch of experiments. Parallel samples were also analyzed to assess precision, with the relative deviation between the parallel and target samples maintained at ≤ 30%. After collection, all samples were stored in the dark at low temperatures and analyzed within one week to minimize chemical loss."

4. Section 2.2 could be divided into four sub-sections for more clearly.

**Response:**

Thank you for your review. We fully agree with your suggestion to improve the structure of the 2.2 section.

We have reorganized section 2.2 into four numbered sub-sections (2.2.1 to 2.2.4) with clear headings, replacing the previous "part 1 to part 4" format. The updated sub-sections are as follows:

"2.2.1 Calculation of the initial concentration of VOCs
2.2.2 Quantifying radical-specific contributions to AOC
2.2.3 Source-resolved AOC sensitivity and attribution framework
2.2.4 Workflow of the AOCPT method"

5. Section 2.2 Part 1: Please define the meaning of in Equation (S3) and provide the data sources for and . It would also be better if the authors could assess the uncertainty of the calculated initial concentration data using this method.

**Response:**

Thank you for your review. We appreciate your detailed attention to the methodology.

Based on the reference (Wang et al., 2022) and our calculation logic, we have revised Text S3 in the Supplementary Materials to provide explicit definitions, data sources, and an uncertainty assessment. The updated sub-sections are as follows (after Equation S5):

"The estimation of the photochemical age ($\Delta t$), which represents the effective reaction time of VOCs with OH· radicals, follows the methodology established by Wang et al. (2022) (Wang et al., 2022b). This approach integrates physical transport processes with chemical reaction kinetics and distinguishes between two meteorological scenarios based on a wind speed threshold of 0.2 m·s$^{-1}$.

Under calm wind conditions (wind speed < 0.2 m·s$^{-1}$), observed VOCs are attributed to local accumulation rather than long-range transport. In this scenario, $\Delta t$ is determined solely by the chemical lifespan ($t_a$), which is defined as the time required for a specific species (assuming an initial concentration of 1 ppbv) to be 80% depleted by OH· radicals. Conversely, under transport conditions (wind speed > 0.2 m·s$^{-1}$), VOCs are considered to originate from upwind pollution sources. Here, $\Delta t$ is calculated as the minimum of the maximum potential photochemical duration ($t_d$) and the physical residence time during transport ($t_r$), as shown in Eq. (S3). A specific exception applies to isoprene; due to its rapid reactivity and emission from biogenic area sources, its age is constrained by the minimum of its chemical lifespan ($t_a$) and transport time ($t_r$).

Regarding the parameters in Eqs. (S3-S5): td denotes the monthly average daytime duration (identified as approximately 13 hours in this study), serving as the upper limit for photochemical processing. The physical residence time ($t_r$) is derived from the average distance to the source ($\overline{D}$) and the monthly average wind speed ($\bar{v}$), adjusted

by a probability weighting function p. As shown in Eq. (S5), p is calculated based on the ratio of the measured monthly dominant wind direction $\overline{WD}$ to the averaged monthly wind direction $WD_{all}$.

For the kinetic calculations, the reaction rate constants ($k_i$) were obtained from the Master Chemical Mechanism (MCM v3.3.1) and the database by Atkinson and Arey (2003) (Atkinson, 2003). The OH· radical concentrations were derived from the 0-D observation-based box model (F0AM) simulations constrained by in-situ measurements.

The uncertainty of the calculated initial VOCs (InVOCs) concentrations primarily arises from three sources: (1) measurement errors of ambient VOCs; (2) uncertainties in reaction rate constants $k_i$; and (3) variations in the simulated [OH·] and estimated Δt. Using the error propagation method, the combined average uncertainty for the calculated initial concentrations is approximately 24%. This level of uncertainty is consistent with similar studies and does not alter the relative identification of dominant species and sources (Niu et al., 2024b; Wang et al., 2022b)."

Atkinson, R.: Kinetics of the gas-phase reactions of OH radicals with alkanes and cycloalkanes, Atmos. Chem. Phys., 3, 2233-2307, 10.5194/acp-3-2233-2003, 2003.

Niu, Y. Y., Yan, Y. L., Xing, Y. R., Duan, X. L., Yue, K., Dong, J. Q., Hu, D. M., Wang, Y. H., and Peng, L.: Analyzing ozone formation sensitivity in a typical industrial city in China: Implications for effective source control in the chemical transition regime, Sci. Total Environ., 919, 10, 10.1016/j.scitotenv.2024.170559, 2024b.

Wang, Z. Y., Shi, Z. B., Wang, F., Liang, W. Q., Shi, G. L., Wang, W. C., Chen, D., Liang, D. N., Feng, Y. C., and Russell, A. G.: Implications for ozone control by understanding the survivor bias in observed ozone-volatile organic compounds system, Npj Climate and Atmospheric Science., 5, 9, 10.1038/s41612-022-00261-7, 2022b.

6. Section 2.2 Part 3: 81 compounds were analyzed during field measurement, but only 38 VOCs were input into the PMF model. Please explain the selection principles. To what extent does the smaller number of compounds used in source apportionment affect the accuracy of analyzing AOC sources?

**Response:**

Thank you for your review. We appreciate this thoughtful comment regarding data selection and model accuracy. The selection principles and evaluated the impact on accuracy have been clarified in Text S9. The revisions are as follows:

"The reduction from 81 to 38 species was strictly based on data quality and model stability requirements (US EPA PMF 5.0 guidelines). We excluded species with low detection frequencies, poor signal-to-noise ratios (S/N < 1), or those that lacked valid source-tracer indications. This ensures the PMF model generates robust and mathematically convergent solutions.

Quantitatively, the 38 selected species accounted for 76% of the total VOCs concentration on average. More importantly, simulation results indicate that these input species capture an average of approximately 96% of the total AOC calculated using the full dataset of 81 species. The typical VOC tracers essential for identifying emission sources in the receptor model were fully retained. Further details are provided in the source identification section. In particular, the box model simulations in the manuscript highlighted specific precursors that drive radical generation and AOC. As illustrated in Fig. 4, species such as trans-2-butene make significant contributions to the Radiation Equivalent Oxidation Capacity (REOC). These critical high-reactivity species were all preserved in the PMF input. Therefore, our assessment confirms that the species screening process has a minimal impact on the results. The accuracy of AOC source apportionment remains well within an acceptable range."

7. Line 329-331: Please explain the reasons.

**Response:**

Thank you for your review. The explanation for the observed high AOC levels have been added as suggested. In the revised manuscript, we clarified that this phenomenon is driven by the Changzhi city's specific industrial characteristics and photochemical conditions. The revisions are as follows (Line 307-309):

"As a typical industrial city characterized by energy and heavy industries (Text S1), it emits substantial amounts of reactive precursors that serve as abundant fuel for photochemistry."

8. What is the reason for the inconsistent units of the vertical axes in the three sub-figures shown in Figure 3? Please make it uniform if possible.

**Response:**

Thank you for your review. We sincerely apologize for this oversight in the figure preparation. We acknowledge that the inconsistent units in the original figure caused confusion. We have re-verified the underlying data and unified the units across all sub-figures in Figure 2 (previous Figure 3) to ensure consistency and comparability. The corrected Figure 2 with uniform vertical axes has been replaced in the revised manuscript. The revisions are as follows:

[Figure]

Fig. 2. Diurnal patterns of AOC simulated

9. In terms of the AOC sensitivity analysis, vehicle and industrial emissions are the prioritized control sources. Whether these sources are also important for O3 and SOA formation? Please clarify it.

**Response:**

Thank you for your review. We appreciate this insightful question. The analysis confirms that industrial processes and vehicle emissions are also critical for $O_3$ and SOA formation, but their importance is often underestimated by traditional metrics. Detailed clarification is as follows:

As shown in Fig. 8, the sensitivity analysis values (RIR) for industrial processes and diesel vehicles were positive for all three parameters (AOC, $O_3$, and SOA), indicating that these sources are indeed the common dominant contributors to both atmospheric oxidation capacity and secondary pollutant formation. Specifically, $O_3$ demonstrates the highest source sensitivity to industrial process (0.029). SOA demonstrates the highest source sensitivity to diesel vehicle emission (0.035), followed by gasoline vehicle emission (0.033) and industrial process (0.030).

However, the key finding is that the emission sources sensitivity magnitude for $O_3$ and SOA is significantly lower than for AOC. Specifically, $O_3$ sensitivity analysis underestimates the contributions of industrial processes (0.029) and diesel vehicle emissions (0.019) by 28.7% and 48.5%, respectively. Similarly, the self-reaction sensitivity analysis underestimates industrial processes (0.030) and diesel vehicle emissions (0.035) by 25.7% and 5.6%, respectively. Meanwhile, compare with the sensitivity analysis of AOC, both $O_3$ and SOA overestimates gasoline vehicles by 48.8% and 172%, respectively.

This implies that while these sources are important for $O_3$ and SOA, relying solely on $O_3$ or SOA targeted sensitivity analysis fails to fully capture their contribution to the oxidative cycle. This may lead to deviations in management policies. Previous studies have shown that AOC quantifies secondary pollutant formation potential (Yu et al., 2022). The AOC-based approach provides a more sensitive and comprehensive indicator for identifying these key drivers.

We have clarified a specific discussion on this point in section 3.3.3 (Line 543-545). The revisions are as follows:

"These results confirm that industrial processes and vehicle emissions are indeed the common dominant drivers for AOC, $O_3$, and SOA formation (as indicated by the positive sensitivity values in Fig. 8)."

Yu, S., Wang, S., Xu, R., Zhang, D., Zhang, M., Su, F., Lu, X., Li, X., Zhang, R., and Wang, L.: Measurement report: Intra- and interannual variability and source apportionment of volatile organic compounds during 2018–2020 in Zhengzhou, central China, Atmos. Chem. Phys., 22, 14859-14878, 10.5194/acp-22-14859-2022, 2022.

10. Species inputs in PMF and F0AM are all in the gas phase, whereas SOA is an aerosol component. Please clarify how you calculated the source sensitivities of SOA.

**Response:**

Thank you for your review. We sincerely apologize for the confusion caused by the lack of clarity in our original description. We thank the reviewer for raising this critical question regarding the link between gas-phase inputs and aerosol outputs.

Specifically, since the F0AM model focuses on gas-phase kinetics, we employed the self-reaction rates of peroxy radicals (Self-rxns) as a proxy to characterize SOA formation potential. As detailed in Text S9, Self-rxns represent the sum of reaction rates for $RO_2 \cdot + HO_2 \cdot$, $RO_2 \cdot + HO_2 \cdot$ and $HO_2 \cdot + HO_2 \cdot$. Previous studies have established that these pathways produce low-volatility oxygenated organic compounds that serve as the primary material for SOA nucleation and growth (Niu et al., 2024a; Lyu et al., 2022). By inputting the PMF-derived source contributions (gas-phase VOCs) into the model, we simulated the changes in these radical reaction rates. The sensitivity of the Self-rxns rate to source changes was then interpreted as the sensitivity of SOA formation potential.

In the revised section 3.3.3, we have explicitly clarified that our approach quantifies the SOA formation potential (via radical self-reactions) rather than the direct physical mass of aerosols. The revisions are as follows (Line 522-530):

"This study further analyzes source sensitivities of AOC, $O_3$, and secondary organic aerosols (SOA) to precursors (Fig. 8). Although the F0AM model simulates gas-phase chemistry, the formation potential of SOA can be effectively characterized by the reaction rates of peroxy radical self-reactions (Self-Rxns: $RO_2 \cdot + HO_2 \cdot$, $RO_2 \cdot + HO_2 \cdot$ and $HO_2 \cdot + HO_2 \cdot$). These reactions typically generate low-volatility compounds (e.g., organic peroxides and accretion products) that readily partition into the particle phase to form SOA (Lyu et al., 2022). Therefore, the sensitivity of Self-Rxns to precursor emissions serves as a robust proxy for SOA source sensitivity. The detailed calculation method of self-reaction is provided in Text S10."

Lyu, X., Guo, H., Zou, Q. L., Li, K., Xiong, E. Y., Zhou, B. N., Guo, P. W., Jiang, F., and Tian, X. D.: Evidence for Reducing Volatile Organic Compounds to Improve Air Quality from Concurrent Observations and In Situ Simulations at 10 Stations in Eastern China, Environ. Sci. Technol., 9, 10.1021/acs.est.2c04340, 2022.

Niu, Y. Y., Yan, Y. L., Dong, J. Q., Yue, K., Duan, X. L., Hu, D. M., Li, J. J., and Peng, L.: Evidence for

sustainably reducing secondary pollutants in a typical industrial city in China: Co-benefit from controlling sources with high reduction potential beyond industrial process, J. Hazard. Mater., 478, 10, 10.1016/j.jhazmat.2024.135556, 2024a.

11. Please supplement the scope of application, limitations, or sources of uncertainty for the AOC approach proposed in this paper.

**Response:**

Thank you for your review. We are grateful for this suggestion to enhance the scientific rigor of our study.

Firstly, in Step 1 of Section 2.2.2, we clarified the inherent limitations of the box model simulation. The revisions are as follows (Line 158-160):
"It should be noted that the F0AM is a zero-dimensional (0-D) box model, which focuses on chemical mechanisms while simplifying physical transport processes."

Secondly, in Section 2.2.4, we elaborated on the unique characteristics and the specific scope of application for the AOCPT method. The revisions are as follows (Line 254-259):
"Compared to existing studies that rely on individual secondary pollutants, the AOCPT method prioritizes the perspective of secondary pollutant formation through quantitative and qualitative analysis. It is primarily applicable to observation-based diagnoses of complex air pollution in urban environments across different seasons. Therefore, this approach provides a robust methodological basis and research direction for the synergistic control and management of secondary pollutants."

Finally, within the Conclusion, we explicitly summarized the method's limitations and primary sources of uncertainty to ensure a balanced perspective. The revisions are as follows (Line 626-631):
"It is primarily applicable to observation-based diagnoses of complex air pollution in urban environments. However, we acknowledge that the 0-D box model assumption simplifies physical transport. The assessment of SOA utilizes gas-phase self-rxns of low-volatility compounds rather than physical mass. Furthermore, uncertainties in this method mainly stem from measurement errors and variations in chemical kinetic constants."

12. The English could be improved to more clearly express the research. For example, (1) line 288: 30% higher; (2) line 289-292: delete "undervalued the concentration of alkene"; (3) line 292-293: delete "the VOCs consumed to"; (4) line 354-356: also the reason why; (5) line 362-365: rewrite these two sentences; (6) line 424: This study used the REOC concept to unify; (7) line 425-428: this sentence was confused; (8) line 438: what was the difference between "REOC metric" and "REOC concept"; (9) line 459-465: re-describe the VOC source apportionment results; (10) Collectively? (11) line

473: delete "which the location of the research case"; (12) line 663 and 667: Critically and crucially? (3) too much "Especially".

**Response:**

We sincerely appreciate your patience and meticulous corrections regarding the language and expression. We fully agree that precise language is essential for clear scientific communication. We have carefully addressed all the specific examples raised by the reviewer and have further conducted a thorough proofreading of the entire manuscript to improve the English quality. The revisions are as follows:

(1) Line 269-270: Corrected to "...was 30.0% higher than MVOCs..."

(2) Line 270-272: Revised to " Specifically, isoprene and anthropogenic alkenes were significantly underestimated by 34.8% and 29.9%, respectively, due to their rapid photochemical depletion" to remove the awkward phrasing.

(3) Line 273: Deleted "the VOCs consumed to" and revised to "...which participated in atmospheric photochemical reactions."

(4) Line 330: Revised "That's also why" to "This explains why..." to be more formal.

(5) Line 338-340: These sentences have been rewritten for better flow: "However, the emission sources and speciation of InVOCs are complex. Therefore, it is crucial to track and identify key VOC species that significantly impact AOC through radical chemistry."

(6) Line 396-397: Revised to: "This study utilized the REOC concept (Eq. (3)) to unify the quantification of InVOCs contributions to radical generation (Fig. 4)."

(7) Line 397-400: We have clarified this sentence: "Given the dominance of OH·-related reactions in AOC (as also shown in 3.2.1), REOC normalizes the capacity of InVOCs to generate various radicals into an equivalent OH· generation capacity. This metric thus serves as an indirect indicator of InVOCs contributions to AOC."

(8) Line 410: We have unified the terminology. We use "REOC concept" when introducing the theoretical framework and "REOC metric" for the calculation tool and results to avoid confusion (such as line 248 and 410).

(9) Line 430-436: We have rewritten the source apportionment description to be more concise and standard: "During the sampling period (Fig. 5a), InVOCs were predominantly contributed by diesel vehicle emissions (26.3%), gasoline vehicle emissions (25.3%), and industrial processes (18.0%). In pollution episodes (Fig. 5b), the contribution from diesel vehicles rose to 30.7%, becoming the dominant source, followed by industrial processes (20.6%) gasoline vehicles emission (23.8%). Notably, contributions from diesel vehicles and industrial processes were 11.5% and 6.8% higher, respectively, during pollution periods compared to clean periods."

(10) Line 440: "Collectively" have been Changed to "Combined, diesel vehicle emissions..." for better accuracy.

(11) Deleted the redundant phrase "which the location of the research case."

(12) Line 643 and 647: We have varied the transition words. "Critically" was

retained in Line 643, while "Crucially" in Line 647 was replaced with "Furthermore" to avoid repetition.

(13) We have checked the entire manuscript and replaced repetitive uses of "Especially" with synonyms such as "Particularly," "Notably," to enhance lexical variety. Such as in line 292, 328, 460 and 471.

---

## Author Comment (AC2)

**Detailed Response to Reviewers' comments**

**Manuscript Number:** egusphere-2025-4355

**Manuscript Title:** An Observation-Based Methodology and Application for Future Atmosphere Secondary Pollution Control via an Atmospheric Oxidation Capacity Path Tracing Approach

**Note:** The Changes of reviewer's suggestion in the revision manuscript were indicated by the red font.

**Reviewer #2:** The manuscript of "An Observation-Based Methodology and Application for Future Atmosphere Secondary Pollution Control via an Atmospheric Oxidation Capacity Path Tracing Approach" presents a novel "Atmospheric Oxidation Capacity Path Tracing" (AOCPT) approach to address the persistent challenge of synergetic $O_3$ and $PM_{2.5}$ control in industrial regions. The methodology is scientifically sound, and the manuscript is generally well-written. But before publication, some details need to be explained. Overall, it is recommended to make minor modifications.

**Response:**

We thank the reviewer for the positive comments. For the comments from reviewers, we have revised the manuscript as better as we can. Please find our point-to-point responses below.

1. Can this AOCPT approach be directly applied to other seasons (e.g., winter, when AOC is lower) or non-industrial urban areas?

**Response:**

We appreciate this insightful question regarding the generalizability of our method. The AOCPT approach is designed as a universal framework based on fundamental physicochemical mechanisms, making it applicable to diverse seasons and urban typologies. The reasons are as follows:

(1) The chemical mechanism of AOCPT approach has universality. The core of the AOCPT approach (the OBM module) relies on the Master Chemical Mechanism (MCM 3.3.1), which explicitly describes the reaction kinetics of thousands of species (Jenkin et al., 2015; Wolfe et al., 2016). Since kinetic rate constants (k) are functions of temperature and pressure, the model inherently accounts for seasonal variations (e.g., lower reaction rates in winter) without requiring structural changes. Previous studies have successfully applied MCM-based box models to investigate pollution formation mechanisms across different seasons R(Yan et al., 2025; Yang et al., 2024).

(2) The source apportionment component of the AOCPT approach possesses inherent adaptability. Its core relies on the Positive Matrix Factorization (PMF) model, which is fundamentally a data-driven technique. Consequently, when applied to diverse scenarios, such as non-industrial urban areas or winter seasons, the input species and resolved source profiles naturally shift. The AOCPT framework seamlessly integrates

these variations, enabling the precise tracing of dominant oxidation drivers specific to any target environment.

(3) The AOCPT approach focus on relative contribution and sensitivity. The AOCPT approach prioritizes the relative contribution and sensitivity of precursors over absolute concentrations (Zheng et al., 2021; Yang et al., 2024). Even in scenarios with lower absolute AOC levels, such as during winter or in non-industrial areas, the 'bottleneck' driving secondary pollution formation still persists. By utilizing the RIA and REOC metrics, this method effectively identifies these limiting factors within constrained oxidation pathways, enabling the formulation of precise control strategies regardless of the overall pollution magnitude.

We have added a describe on this applicability in line 256-257 of the revised manuscript.

"It is primarily applicable to observation-based diagnoses of complex air pollution in urban environments across different seasons."

Jenkin, M. E., Young, J. C., and Rickard, A. R.: The MCM v3.3.1 degradation scheme for isoprene, Atmos. Chem. Phys., 15, 11433-11459, 10.5194/acp-15-11433-2015, 2015.

Wolfe, G. M., Marvin, M. R., Roberts, S. J., Travis, K. R., and Liao, J.: The Framework for 0-D Atmospheric Modeling (F0AM) v3.1, Geosci. Model Dev., 9, 3309-3319, 10.5194/gmd-9-3309-2016, 2016.

Yan, Y. L., Niu, Y. Y., Duan, X. L., Yue, K., Dong, J. Q., Yang, C., Hu, D. M., Wang, Y. H., Li, J. J., and Peng, L.: Insight into carbonyl source based on improved source apportionment method: Alkene regulate secondary formation, J. Hazard. Mater., 489, 11, 10.1016/j.jhazmat.2025.137649, 2025.

Yang, J., Zeren, Y., Guo, H., Wang, Y., Lyu, X., Zhou, B., Gao, H., Yao, D., Wang, Z., Zhao, S., Li, J., and Zhang, G.: Wintertime ozone surges: The critical role of alkene ozonolysis, Environmental Science and Ecotechnology., 22, 100477, https://doi.org/10.1016/j.ese.2024.100477, 2024.

Zheng, Y., Chen, Q., Cheng, X., Mohr, C., Cai, J., Huang, W., Shrivastava, M., Ye, P., Fu, P., Shi, X., Ge, Y., Liao, K., Miao, R., Qiu, X., Koenig, T. K., and Chen, S.: Precursors and Pathways Leading to Enhanced Secondary Organic Aerosol Formation during Severe Haze Episodes, Environ. Sci. Technol., 55, 15680-15693, 10.1021/acs.est.1c04255, 2021.

2. For REOC parameters, the conversion efficiencies α and β are critical. Please provide the screening criteria for the reaction pathways used and perform a sensitivity analysis to show how variations in these parameters affect the overall AOC conclusions.

**Response:**

Thank you for your review. We appreciate this critical question regarding the robustness of our metric. We have addressed this comment from screening criteria, uncertainty assessment and sensitivity analysis. The detailed revisions are included in the new Text S4 of the supplementary materials. Accordingly, we have added a cross-reference in the main manuscript (Line 187-189):

"Sensitivity analysis confirms that the reliability of identifying key reactive species was not compromised by parameter uncertainties (detailed in Text S4 and Fig.

S1)."

(1) Screening criteria. The essence of REOC is to indirectly unify the quantification of atmospheric oxidation capacity (AOC) by normalizing various radicals to an equivalent OH· radical generation capacity. As detailed in the manuscript (Section 2.2.2 and 3.2.1), OH· is the primary driver of AOC. Therefore, the screening criteria for reaction pathways were based on the ROx· radical cycling mechanism. We selected pathways that effectively convert intermediate radicals (HO2· and RO2·) back into the OH· pool. Parameters $\alpha$ and $\beta$ represent the efficiency of these recycling channels.

(2) Uncertainty assessment. We quantified the uncertainty of REOC results based on the propagation of errors from input parameters (measurement errors, kinetic constants, and model-derived results). Consequently, the derived values for $\alpha$ and $\beta$ are $0.7\pm0.1$ and $4.0\pm0.8$, respectively. This corresponds to an estimated uncertainty of approximately 20%, which falls well within the typical uncertainty range (10% to 30%) for key atmospheric chemical reactions reported in existing studies (Deguillaume et al., 2007).

(3) Sensitivity analysis. We performed a sensitivity test by varying $\alpha$ and $\beta$ by $\pm20\%$ (Niu et al., 2024). This range was chosen because it covers the typical uncertainty range of kinetic rate constants (10% to 30%) and measurement errors in atmospheric chemistry (Deguillaume et al., 2007; Niu et al., 2024). The results (Fig. S1) confirm that while absolute values fluctuate, the relative ranking of dominant species (e.g., trans-2-butene) remains unchanged, proving the robustness of our conclusions.

Deguillaume, L., Beekmann, M., and Menut, L.: Bayesian Monte Carlo analysis applied to regional- scale inverse emission modeling for reactive trace gases, #N/A, 112, 13, 10.1029/2006jd007518, 2007.

Niu, Y. Y., Yan, Y. L., Xing, Y. R., Duan, X. L., Yue, K., Dong, J. Q., Hu, D. M., Wang, Y. H., and Peng, L.: Analyzing ozone formation sensitivity in a typical industrial city in China: Implications for effective source control in the chemical transition regime, Sci. Total Environ., 919, 10, 10.1016/j.scitotenv.2024.170559, 2024.

3. Whether the contents in similar color boxes in Part 2 and Part 3 were similarly meaning? if yes, added legend in fig.1. otherwise, distinguish in differ box colors.

**Response:**

We thank the reviewer for the careful observation regarding the visualization in Figure 1. The boxes in Part 2 and Part 3 belong to different methodological modules (OBM and PMF) and do not represent identical categories. To eliminate any potential ambiguity, we have followed your suggestion and adjusted the color scheme in the revised Figure 1. We now use distinct box colors to visually differentiate the workflows

of the OBM module (Part 2) and the PMF module (Part 3). This modification ensures that the distinct logical steps within each module are clearly separated and easily understood.

[Figure]

Fig. 1. The workflow of the AOCPT method.

4. The descriptions of Fig. S in manuscript were incorrect, and check them throughout manuscript.
**Response:**

We sincerely apologize for the oversight regarding the descriptions of Supplementary Figures.

We have conducted a thorough cross-check of the entire manuscript against the supplementary materials. The formatting of these citations throughout the manuscript have been standardized. Additionally, we have corrected all mismatches to ensure that every citation in the main text corresponds accurately to the figures and captions in the Supplementary Information.

5. For better clarity, please label the sub-figures using letters such as (a), (b), etc. Additionally, the bar chart showing the contributions of different oxidizing agent pathways should not be parallel to the time axis. I suggest replacing this bar chart with a pie chart to improve readability.
**Response:**

We appreciate your helpful suggestion to enhance the visual clarity of the results.

We have redesigned Figure 2 in the revised manuscript to address these points:

(1) We have explicitly labeled the sub-figures as (a) Total, (b) Pollution period, and (c) Clean period for easier referencing.

(2) As suggested, we have incorporated pie charts (donut charts) within each panel to represent the overall contribution proportions of the three major oxidants (OH·, $O_3$ and $NO_3$·). This replaces the previous layout and provides a more intuitive visualization of the dominant oxidation pathways under different conditions.

The revisions are as follows:

[Figure]

Fig. 2. Diurnal patterns of AOC simulated

6. The manuscript contains an excessive number of figures. I recommend moving some figures to the Supporting Information (SI), such as Fig. 2 and Fig. 4 etc.
**Response:**

We appreciate your suggestion to streamline the manuscript.

We fully agree that moving descriptive figures to the Supplementary Information improves the readability and flow of the main text.

We have moved the original Figure 2 (Diurnal variations of atmospheric pollutants) and Figure 4 (Diurnal variations of free radicals) to the Supplementary Materials. They are now labeled as Figure S3 and Figure S6, respectively.

Renumbering: Accordingly, we have renumbered all subsequent figures in the main manuscript. For instance, the original Figure 3 (Diurnal patterns of AOC) has become the new Figure 2, and the original Figure 5 (ROx radical cycle) has become the new Figure 3.

Citation Updates: All citations and references to these figures in the main text have been updated to correspond with the new numbering system.

7. The manuscript contains many long and complex sentences, which significantly hinder readability. I suggest the authors revise the text by breaking down complex structures into shorter, more reader-friendly sentences to ensure the scientific findings are communicated clearly.
**Response:**

We sincerely appreciate your valuable suggestion regarding the readability of the manuscript. We fully agree that clear and concise expression is essential for effectively communicating scientific findings. We have carefully reviewed the entire manuscript and made extensive revisions to simplify sentence structures.

Specifically, we have systematically broken down complex, multi-clause sentences into shorter, independent sentences to improve logical flow and clarity. Special attention was paid to the Abstract and Introduction sections, where the sentence structures were significantly optimized to ensure the research background and objectives are presented clearly. We have also refined the language throughout the Results and Discussion sections to avoid ambiguity.